# The Pathophysiology and the Management of Radiocontrast-Induced Nephropathy

**DOI:** 10.3390/diagnostics12010180

**Published:** 2022-01-12

**Authors:** Eunjung Cho, Gang-Jee Ko

**Affiliations:** Division of Nephrology, Department of Internal Medicine, Korea University Medical Center, Seoul 08308, Korea; icdej@naver.com

**Keywords:** acute kidney injury, contrast-induced nephropathy, risk factor, pathogenesis, oxidative stress, prevention

## Abstract

Contrast-induced nephropathy (CIN) is an impairment of renal function that occurs after the administration of an iodinated contrast medium (CM). Kidney dysfunction in CIN is considered transient and reversible in most cases. However, it is the third most common cause of hospital-acquired acute kidney injury and is associated with increased morbidity and mortality, especially in high-risk patients. Diagnostic and interventional procedures that require intravascular CM are being used with increasing frequency, especially among the elderly, who can be particularly susceptible to CIN due to multiple comorbidities. Therefore, identifying the exact mechanisms of CIN and its associated risk factors is crucial not only to provide optimal preventive management for at-risk patients, but also to increase the feasibility of diagnostic and interventional procedure that use CM. CM induces kidney injury by impairing renal hemodynamics and increasing the generation of reactive oxygen species, in addition to direct cytotoxicity. Periprocedural hydration is the most widely accepted preventive strategy to date. Here, we review the latest research results on the pathophysiology and management of CIN.

## 1. Introduction

Contrast-induced nephropathy (CIN) is an impairment of kidney function that occurs after the administration of iodinated contrast medium (CM). It is the third most common cause of hospital-acquired acute kidney injury (AKI) and is associated with prolonged hospital stay and increased morbidity and mortality [1,2,3]. The reported incidence of CIN varies from <1% to greater than 50% depending on patient risk factors, type of procedure, and definition of CIN [4,5,6,7]. Diagnostic and interventional procedures that require intravascular CM are being used with increasing frequency, especially among the elderly, who can be particularly susceptible to CIN due to multiple comorbidities such as chronic kidney disease (CKD) and diabetes mellitus. Therefore, it is important to understand the precise risks and pathophysiology of CIN to provide optimal preventive management. We review the latest research on the pathophysiology and management of CIN. For this review, a literature search was performed using electronic databases such as PubMed and Embase. The search strategy included any original article or review about contrast-induced nephropathy. We used combinations of the following search terms: contrast, iodinated contrast media, nephropathy, risk, score, incidence, guideline, definition, intravenous, intra-arterial, biomarker, chronic kidney disease, diabetes, metformin, angiotensin, pathophysiology, oxidative stress, Rho, ROCK, sirtuin, SIRT1, Nrf2, NLRP3 inflammasome, prevention, hydration, RenalGuard system, sodium bicarbonate, N-acetylcysteine, and statin. Both in vivo and in vitro experimental studies and clinical studies were reviewed.

## 2. Terminology and Definition

The term used to describe kidney injury following exposure to CM has changed over time. The most recent guidelines from the American College of Radiology (ACR) Committee on Drugs and Contrast Media suggested contrast-associated acute kidney injury (CA-AKI), formerly called post-contrast acute kidney injury (PC-AKI), as a general term to describe a decline in kidney function that occurs within 48 h after the intravascular administration of iodinated CM [8]. CA-AKI is a correlative diagnosis, regardless of the exact etiology of AKI. On the other hand, they suggested the term contrast-induced acute kidney injury (CI-AKI), formerly known as CIN, as a causative diagnosis describing AKI due to CM.

However, it is difficult to identify the definite cause of AKI in patients who undergo a procedure using CM because various patient- and procedure-related factors can influence kidney function, such as hemodynamic instability or atheroembolism caused by catheter manipulation. The terms used in previous clinical studies have been inconsistent, and it is difficult to differentiate CI-AKI from CA-AKI in most studies, mainly due to the lack of a suitable control group. The incidence of CI-AKI might have included cases of CA-AKI, although CI-AKI is a subgroup of CA-AKI. We use the traditional term CIN in this article because we discuss the pathophysiology and management of kidney injury caused by CM.

Kidney dysfunction in CIN is usually reversible; the decline in kidney function occurs 2–3 days after exposure to CM and returns to the baseline level within 1–2 weeks [9,10]. The diagnostic criteria for CIN have changed in the same manner as those for AKI, irrespective of etiology [4].The most recent definition from the Kidney Disease Improving Global Outcomes (KDIGO) initiative diagnoses CIN if one of the following occurs within 48 h after intravascular administration of CM: (1) absolute increase in serum creatinine (sCr) ≥ 0.3 mg/dL (≥26.4 μmol/L), (2) relative increase in sCr ≥ 50% (≥1.5 times baseline), or (3) urinary volume <0.5 mL/kg/h for ≥6 h, which is now adopted as the standard for both CA-AKI and CI-AKI [11]. The European Renal Best Practice working group, the Contrast Media Safety Committee (CMSC) of the European Society of Urogenital Radiology (ESUR), and the ACR Committee on Drugs and Contrast Media all recommend using the KDIGO definition for CIN [8,10,12]. The CMSC suggested events within 48–72 h after exposure to CM as a practical definition [10].

Diverse definitions of kidney injury after CM administration have been used in clinical studies: an absolute increase in sCr ≥0.3–0.5 mg/dL or a relative increase in sCr ≥ 25–50% from baseline values. A relative increase ≥25% is the most sensitive indicator, and an absolute increase ≥0.5 mg/dL is the least sensitive [4,13,14]. Therefore, the incidence of CIN reported in clinical studies should be interpreted carefully in light of the definition used.

Although AKI is defined as a change in sCr, the elevation of sCr might not be a sensitive marker for assessing changes in glomerular filtration rate (GFR). SCr increases slowly after a reduction in GFR because creatinine is distributed in the total body water. Furthermore, creatinine is secreted by kidney tubular cells, and the sCr value is affected by multiple factors such as muscle mass, age, sex, and hydration status [15,16]. Because of the low sensitivity and specificity of sCr, various new biomarkers have been studied to detect kidney injury more precisely after CM administration [17,18]. Carmen et al. reviewed biomarkers of CIN and divided them into two categories: (1) functional biomarkers that can detect a decrease in kidney function with more sensitivity than creatinine, including cystatin C, and (2) structural kidney damage biomarkers such as neutrophil gelatinase-associated lipocalin, liver-type fatty acid-binding protein, and kidney injury molecule-1 (KIM-1) [15]. Some biomarkers can detect early kidney injury even before functional change develops, and some can predict the occurrence or prognosis of CIN. In a recent sub-study of the PRESERVE trial that evaluated plasma and urine biomarkers, only plasma KIM-1 was significantly associated with CIN [19]. However, for general application of biomarkers as a routine procedure in clinical practice, further studies are needed to evaluate and validate the clinical significance and cutoff values for each one.

## 3. Risk Factors

Because no definite treatment to ameliorate CIN has been established, the importance of preventive measures has been highlighted, and identifying patients at high risk for CIN is the first step in prevention. A variety of risk factors has been reported and can be divided into patient-related and procedure-related risk factors, as summarized in Table 1.

Impaired kidney function, diabetes, advanced age, and preexisting intravascular volume depletion before CM administration are frequently reported patient-related risk factors [20,21]. Female sex, cardiovascular disease, and concomitant use of medications known to be nephrotoxic have also been reported as factors that increase the risk of CIN [16,22,23,24].

Patients with CKD, particularly diabetic kidney disease, are most susceptible to CIN [6,25,26]. Contrast-enhanced computed tomography (CECT) was associated with higher risk of AKI in patients with estimated GFR (eGFR) less than 30 mL/min/1.73 m^2^ (OR 1.36, 95% CI 1.09–1.70, *p* = 0.007), but not in patients with an eGFR greater than 45 mL/min/1.73 m^2^ [27]. Similarly, in cancer patients who underwent CECT using a reduced dose of iodixanol, the incidence of CIN was reported to increase according to eGFR: 4.6% for eGFR 45–60 mL/min/1.73 m^2^, 7.4% for eGFR 30–45 mL/min/1.73 m^2^, and 16.7% for eGFR < 30 mL/min/1.73 m^2^ [28]. Though the underlying mechanism by which diabetes is implicated in an increased incidence of CIN is complex and not fully understood, Li et al. recently summarized it as follows. The pathophysiologic changes that occur in the kidney under high-glucose status in diabetic kidney disease, including enhanced oxygen consumption, increased oxidative stress with reactive oxygen species (ROS) generation, and dysregulation of vasoactive substances, were associated with an increased risk of CIN [29]. The immunological changes and ROS-related signaling pathways of diabetes also overlap with the pathophysiologic processes of CIN in ways that aggravate kidney injury by means of inflammation and apoptosis. The reductions in GFR that occur with increased age are accompanied by increased vascular resistance and decreased nitric oxide (NO) generation, and those conditions predispose elderly people to CIN [30,31,32]. Decreased volume status increases the nephrotoxicity of CM by increasing the concentration of CM on each nephron, increasing the viscosity of blood and urine, and augmenting kidney hypoxia.

Although those patient-related factors are generally accepted as risk factors for AKI regardless of its underlying cause and are not specific for CIN, exposure to more risk factors should be considered as resulting in a higher incidence of CIN. Therefore, preventive measures for modifiable risk factors and caution for CIN should be adopted.

The following procedure-related risk factors are considered to be associated with CIN: type, volume, and route of CM administration, and repetitive CM administration within a short period of time (24–72 h) [33].

First generation CM had significant side effects due to its high osmolality (1000–2000 mOsmol/kg H_2_O) [34]. The development of CM reduced its osmolality to low-osmolar CM (LOCM, second generation, 300–800 mOsmol/kg H_2_O) and iso-osmolar CM (IOCM, third generation, iodixanol, 290 mOsmol/kg H_2_O) [35,36]. High osmolar CM is rarely used today [22,37]. Guidelines recommend the use of either LOCM or IOCM. IOCM has osmolality similar to that of plasma, but it has a higher viscosity, which can limit the benefits from lowering the osmolality compared with LOCM. Results comparing the risk of CIN after using LOCM and IOCM have been controversial [38,39,40,41]. A meta-analysis comparing the risk of CIN between iodixanol (IOCM) and iopromide (LOCM) in eight randomized controlled trials (RCTs) with 3532 patients undergoing coronary angiography (CAG) with or without percutaneous coronary intervention (PCI) found no significant difference in the incidence of CIN, but adverse cardiovascular events were significantly fewer in the iodixanol group than the iopromide group [41]. A retrospective study with a total of 9431 patients who received elective PCI found that IOCM significantly reduced the incidence of CIN compared with LOCM, but the type of CM did not affect 2-year all-cause mortality [39]. A meta-analysis that included diabetic patients demonstrated that IOCM was superior to LOCM when CIN was defined as an absolute sCr increase ≥0.5 mg/dL but not when it was defined as a relative sCr increase ≥25% [38]. Those authors also found that IOCM was associated with fewer adverse events than LOCM. Similarly, in different rat models of CIN, iohexol induced more severe kidney injury than iodixanol [42,43]. However, the recent meta-analysis by Han et al. showed no significant difference between IOCM and LOCM in the incidence of CIN among diabetic patients, although a significant reduction in the risk of CIN was observed in the iodixanol group compared with the iohexol group [40]. On the other hand, Zhang et al. compared the roles of iodixanol and iohexol on endothelial cell dysfunction among patients with cardiovascular disease [44]. The levels of circulating CD31^+^/CD41a^−^ endothelial microparticles (EMPs) and platelet microparticles (PMPs) were higher in patients with diabetes who received intra-arterial (IA) iohexol than in those who received iodixanol or patients without diabetes who received iohexol, whereas CD62E^+^ EMPs decreased significantly in patients receiving iodixanol. In an in vitro experiment using human umbilical vein endothelial cells, iohexol induced the release of more CD31^+^/CD41a^−^ EMPs than iodixanol, which resulted in greater apoptosis, whereas CD62E^+^ EMP levels were lower after exposure to iodixanol than to iohexol. Elevated EMPs and PMPs can lead to an increase in blood viscosity and be associated with inflammation and thrombosis that can contribute to endothelial damage. Given their results, those authors suggested that iodixanol was a better choice than iohexol in patients with diabetes undergoing CAG. IA CM administration is known to be associated with a higher risk of CIN than intravenous (IV) CM administration [4,45,46,47,48]. In the 2018-ESUR guideline, the IA route was further divided into first and second pass kidney exposure to consider the degree of CM dilution caused by circulation before it reaches the kidney arteries [10]. It suggested different cutoff values for eGFR as risk factors for CIN according to the route of CM administration: eGFR < 45 mL/min/1.73 m^2^ for IA injection with first pass kidney exposure vs. eGFR < 30 mL/min/1.73 m^2^ for IV or IA injection with second pass kidney exposure. In addition to more abrupt and concentrated CM exposure to the kidneys with a supra-renal arterial CM injection, atheroembolism related to catheter manipulation also poses an increased risk of CIN during CAG.

It has been shown that a higher volume of CM increases the risk of CIN, especially during CAG [49,50]. Although there is no specific threshold dose of CM, the ESUR guideline advises clinicians to limit the ratio CM dose (in grams of iodine)/absolute eGFR (in mL/min) <1.1 or the ratio CM volume (in mL)/eGFR (in mL/min/1.73 m^2^) <3.0, assuming a CM concentration of 350 mg iodine/mL, for IA administration with first pass kidney exposure [51]. There is an effort to use even less CM during CAG. In a recent study by Gurm et al., ultra-low contrast volume, defined as contrast volume ≤ the patient’s estimated creatinine clearance, was applied in 9857 patients undergoing PCI, and it was associated with a significant reduction in AKI, particularly in high-risk patients [50].

Mehran first developed a risk scoring system, which involves eight clinical and procedural variables, such as age over 75 years, existence of decrease in renal function, hypotension, congestive heart failure, diabetes, anemia, use of intra-aortic balloon pump (IABP), and large contrast volume, to predict CIN after PCI. More than half of the patients with a higher score than 16 were reported to experience CIN. Exposure to more risk factors is valuable to define patients at risk for CIN [21].

## 4. Pathophysiology

The exact pathophysiology of CIN is not fully understood. Direct cytotoxicity, altered intrarenal hemodynamics, and ROS generation have been proposed as the main pathophysiologic mechanisms of CIN [52]. Those three mechanisms influence and aggravate one another, creating a vicious cycle that ultimately leads to inflammation, tubular cell apoptosis, and impaired kidney function (Figure 1).

CM has a direct cytotoxic effect on kidney tubular epithelial cells and vascular endothelial cells [35]. All types of CM showed cytotoxic effects in vitro [52]. CM induced vacuolization in kidney tubular cells by pinocytosis (osmotic nephrosis), mitochondrial dysfunction that led to ROS generation and apoptosis, and endoplasmic reticulum stress that activated intrinsic apoptotic pathways [53]. Loss of the tubular brush border and cell membrane integrity and sloughing of the tubular epithelial cells into the lumen were caused by the direct cytotoxicity of CM [33,54].

CM administration induces transient vasodilation followed by vasoconstriction that can be sustained for several hours in the kidney vasculature as a result of alterations in kidney vasomodulators such as endothelin, adenosine, and NO [53,55]. Vasoconstriction of afferent arterioles reduces GFR and kidney blood flow, causing kidney parenchymal hypoxia [56]. The kidney outer medulla is in a relative hypoxic situation because of tubular ion transport in the basal state and the low partial pressure of oxygen with limited blood flow caused by the unique anatomy of the kidney vasculature. Hence, the thick ascending limbs of the loop of Henle (TAL) and segments of the proximal kidney tubules in the outer medulla are particularly susceptible to hypoxic injury [57].

The high osmolality and viscosity of CM causes osmotic diuresis, an increase in tubular pressure, and decrease in tubular and blood flow rates, all of which lead to an increase in tubular oxygen demand and a decrease in kidney blood supply [58]. Furthermore, CM induces direct vasoconstriction of the vasa recta through endothelial dysfunction and changes in red blood cell structure and function, both of which worsen kidney medullary hypoxia [34,59]. This mismatch between the metabolic demands of the TAL and the kidney medullary blood supply leads to oxidative stress. Tubular transport is associated with ROS production and the dense mitochondrial population in the proximal tubule and TAL is the major source of ROS [52]. Moreover, CM retention in the tubular lumen caused by the decreased tubular flow rate augments its cytotoxic impact. Ischemic and cytotoxic tubular cell damage again induces tubuloglomerular feedback, which enhances vasoconstriction of the afferent arteriole and produces further decreases in kidney blood flow and GFR [60].

The increase in ROS generation after exposure to CM has been observed in various in vitro and in vivo studies and can be explained partly by the diminished availability or activity of cellular antioxidant systems [61,62]. As explained above, both the direct cytotoxic effects of CM on tubular cells and kidney medullary hypoxia caused by vasoconstriction enhance ROS generation. Subsequently, ROS constrict kidney microcirculation and affect kidney vascular tone by modulating vasoactive substances such as NO [63]. In addition, oxidative DNA damage and multiple intracellular signaling pathways related to ROS lead to necrosis or apoptosis of kidney tubular cells. Because ROS is considered to play a central role in the pathogenesis of CIN, current research focuses on its involvement in CIN, either to elucidate the pathogenesis mechanism or to find an effective preventive or therapeutic target.

With regard to the pathophysiologic role of ROS in CIN, Kusirisin et al. reviewed in vitro and in vivo reports from PubMed up to September 2019 [62]. They summarized the intracellular signaling mechanisms associated with ROS in four pathways: (1) the mitogen-activated protein kinase (MAPK) pathway, which includes extracellular signal-related kinases, c-JUN N-terminal kinase, and p38; (2) the silent information regulator 1(SIRT1) pathway, which includes SIRT1, forkhead box type O transcription factors(FoxO), nuclear factor-κB(NF-κB), peroxisome proliferator-activated receptor gamma-assisted activating factor-1(PGC-1), and p53; (3) the Rho/Rho-kinase(Rho/ROCK) pathway, which includes myosin phosphatase target subunit 1 and NF-κB; (4) the nuclear factor erythroid2-related factor 2/heme oxygenase 1(Nrf-2/HO-1) pathway, which includes Nrf-2, nicotinamide adenine dinucleotide phosphate quinone oxidoreductase 1, glutathione, and HO-1.

CM increased ROS generation by upregulating nicotinamide adenine dinucleotide phosphate oxidase 2 (Nox2), Nox4, and p22phox, which led to apoptosis through the MAPK pathway [64,65,66]. ROCK belongs to the AGC (protein kinase A/protein kinase G/protein kinase C) family of serine/threonine kinases, which is a downstream target of the small GTPase Rho [67]. The Rho/ROCK pathway was reported to be activated by ROS [68]. ROCK-2 activity increases in CIN and regulates inflammation in the kidney [69]. Inhibiting the Rho/ROCK pathway decreased inflammation, the intracellular ROS level, and kidney cell apoptosis in mice with CIN and also induced kidney vasodilation and increased kidney artery blood flow [69].

Both the SIRT1-mediated and Nrf2-mediated pathways are involved in renoprotection against CM-induced oxidative stress and kidney cell apoptosis. SIRT1 decreased after exposure to CM, but Nrf2 expression increased during CM-induced oxidative stress as a cytoprotective response [70,71,72,73,74]. Activating either SIRT1 or Nrf2 attenuated CIN via diverse downstream mechanisms.

Sirtuins belong to a conserved family of nicotinamide adenine dinucleotide (NAD^+^)-dependent deacetylases that is involved in multiple cellular functions related to proliferation, DNA repair, mitochondrial energy homeostasis, and antioxidant activity [75]. SIRT1 is the most widely studied sirtuin and is located in the nucleus, where it regulates both nucleosome histone acetylation and the activity of a variety of transcriptional factors and cofactors, including NF-κB, p53, FoxO, hypoxia-inducible factor(HIF)-2α, and PGC-1α [71,76]. Hong et al. examined the protective role of SIRT1 in CIN using NRK-52E cells and mice [71]. Iohexol decreased SIRT1 and PGC-1α expression both in vivo and in vitro. Using resveratrol to activate SIRT1 reduced oxidative stress, inflammation, and tubular cell apoptosis in mouse kidneys and increased the expression of SIRT1, PGC-1α, and dephosphorylated FoxO1 (activated form). Likewise, using siRNA to inhibit SIRT1 accentuated the decrease in NRK-52E cell viability after iohexol treatment. PGC-1α increased mitochondrial superoxide dismutase (SOD2) level and attenuated oxidative stress. Thus, the SIRT1-PGC-1α-Foxo1 signaling pathway was found to play a role in the development of CIN in mice. Wang et al. examined the involvement of the SIRT1–PGC-1α–HIF-1α signaling pathway in CIN using a rabbit model of diabetic nephropathy (DN rabbits) and HK-2 cells [77]. Resveratrol, a SIRT1 activator, inhibited iohexol-induced HK-2 cell apoptosis, which was enhanced by treatment with 2-MeOE2 (a HIF-1α inhibitor) under high-glucose conditions. In DN rabbits, SIRT1 activation was associated with the upregulation of PGC-1α and downregulation of HIF-1α, Bax, cleaved caspase-3, and cytochrome C protein. This was further verified in HK-2 cells under high-glucose conditions via 2-MeOE2 and SIRT1 inhibition using Ex527.

Nrf-2, a transcription factor, stimulates the transcription of genes that encode detoxifying and antioxidant enzymes, and increased Nrf-2 expression was noted as a cytoprotective response after exposure to CM [70,73,78]. Kim et al. evaluated the role of Nrf-2 in CIN using Nrf2 knockout mice and NRK-52E cells [72]. Loss of Nrf-2 function enhanced ROS production, inflammation, and apoptosis after iohexol treatment, whereas Nrf-2 activation via CDDO-Me co-treatment with iohexol attenuated tubular cell injury. Zhou et al. used a rat model of CIN and Nrf2-silenced HK-2 cells to reveal that the protective role of Nrf2 in CIN is mediated by the Nrf2/Sirt3/SOD2 signaling pathway [74]. SIRT3, a NAD^+^-dependent deacetylase localized in the mitochondrial matrix, regulates a variety of cellular processes and maintains mitochondrial function. SIRT3 protects against oxidative stress by transforming acetylated SOD2 into SOD2. Nrf2 activation using tert-butylhydroquinone reduced oxidative stress and kidney injury and increased SIRT3 and SOD2 expression in CIN rats. The Nrf2-mediated SIRT3/SOD2 pathway was validated in vitro. The expression of SIRT3 and SOD2 increased in HK-2 cells but decreased in Nrf2-silenced cells after ioversol treatment.

Apart from those four pathways, ROS-related mechanisms involving the NLRP3 inflammasome have been studied [79,80,81,82,83]. The NLRP3 inflammasome is associated with inflammation and apoptosis during AKI. Tan et al. demonstrated the involvement of the S100A8/A9-TLR4-NLRP3 inflammasome pathway in the development of CIN using rats with CIN and NRK-52E cells [83]. Lin et al. reported that PINK1-Parkin-mediated mitophagy protected kidney tubular epithelial cells by decreasing mitochondrial ROS and inhibiting the NLRP3 inflammasome [84]. Xu et al. revealed that the protective effect against CIN offered by microRNA-30c, which is upregulated under contrast exposure, is mediated by suppression of the NLRP3 inflammasome [85]. Attenuating CIN by directly inhibiting NLRP3 was demonstrated in in vivo (nlrp3 or casp1 knockout mice) and in vitro (treatment with MCC950, a selective NLRP3 inflammasome inhibitor) experiments that also resulted in the upregulation of HIF-1α and BNIP3-mediated mitophagy [82].

## 5. Management

At present, prevention is the best management strategy for CIN and can be divided into patient-, procedure-, and pathophysiology-related methods (Table 2). All patients receiving intravascular CM should be evaluated for the risk of CIN, and clinicians should adopt interventions for modifiable risk factors such as dehydration and consider discontinuing nephrotoxic medications before CM administration. Herein, we review preventive strategies studied in adult patients undergoing procedures using CM. Therefore, the management of CIN in this article refers primarily to the adult population.

Since Mehran first developed a risk scoring system, which involves eight clinical and procedural variables, to predict CIN after PCI, several simpler risk assessment models have been proposed [21,86,87,88,89]. The ACEF (age, creatinine, and ejection fraction) score was originally developed in 2009 to assess the mortality risk in patients undergoing elective cardiac operations, but that simple risk scoring system has subsequently been validated in other clinical conditions, including CIN after CAG or PCI. It is now the basis for comparison, along with Mehran’s score system, for new CIN risk scoring systems [87,88,90,91]. Zeng et al. proposed a risk score based on four variables (age > 75 years, acute myocardial infarction (AMI), sCr > 1.5 mg/dL, use of an IABP), and Ni et al. suggested a pre-procedure risk score that considers five factors (age > 75 years, hypotension, AMI, sCr ≥ 1.5 mg/dL, and congestive heart failure) [87,88]. Those risk scoring systems have been evaluated for their ability to predict CIN, procedure-related mortality, and major adverse clinical events in patients undergoing CAG or PCI. However, those systems are not yet relevant for IV administration of CM or patients receiving non-coronary angiography. External validation of those models and the development of a novel risk scoring system that can be generally applied to all cases of CM use are required.

When clinically feasible, it is recommended to withhold nonessential nephrotoxic medications before CM administration, these are listed in Table 1 [51,92,93]. Renin–angiotensin–aldosterone system (RAAS) blockers (angiotensin-converting enzyme inhibitors (ACEI) and angiotensin receptor blockers (ARB)) are generally used in patients with cardiovascular disease, CKD, and diabetes. Because RAAS blockade can change renal hemodynamics and induce AKI, the effect of ACEI/ARBs on the incidence of CIN is of great concern [94]. Wu et al. performed a meta-analysis with 14 studies composed of 15,447 patients (7288 treated with ACEI or ARB and 8159 in the control group) undergoing CAG [95]. The overall estimate demonstrated significantly increased risk of CIN in the ACEI/ARB group compared to the control group (OR 1.50, 95% CI 1.03–2.18, *p* = 0.03), but the association was not observed in the seven RCTs (OR 0.88, 95% CI 0.41–1.90, *p* = 0.74). In a recent meta-analysis by Wang et al. that included 12 studies with 14 trials, containing 4864 patients (2484 treated with RAAS blockers and 2380 in the control group), the pooled relative risk of CIN incidence in the RAAS blocker group was 1.22 (95% CI 0.81–1.84) [96]. However, an increased risk of CIN in the RAAS blocker group was observed among older people (RR 2.02, 95% CI 1.21–3.36), non-Asians (RR 2.30, 95% CI 1.41–3.76), chronic users (RR 1.69, 95% CI 1.10–2.59), and studies with larger sample sizes (population ≥ 200, RR 1.83, 95% CI 1.28–2.63).

Among studies included in the above described meta-analyses, only a few RCTs directly investigated the effects of withholding ACEI/ARB on the incidence of CIN. Discontinuing captopril 36 h before PCI did not change the incidence of CIN in patients with sCr ≤ 1.5 mg/dL or GFR ≥ 60 mL/min [97]. Withholding ACEI/ARB 24 h before CAG did not appear to influence the incidence of CIN in patients with CKD stages 3–4 [98]. Recently, Motes et al. performed a retrospective study and analyzed changes in renal function during one-month post CAG in CKD stages 2–5 patients who take ACEI/ARB and are not on dialysis [94]. This study revealed that the continuation of ACEI/ARB was not associated with significant renal injury after CAG. However, post-hoc analysis of an RCT by Wolak et al. showed that the continuation of ACEI/ARB was associated with a significant decrease in eGFR 48 h post CAG in patients with baseline eGFR < 60 mL/min compared to the discontinuation group, while there was no significant difference in changes of renal function between the two groups in patients with eGFR ≥ 60 mL/min [99]. Likewise, in patients with moderate renal insufficiency (Cr ≥ 1.7 mg/dL within 3 months and/or Cr ≥ 1.5 mg/dL within 1 week before cardiac catheterization), withholding ACEI/ARB resulted in a non-significant reduction in CIN and a significant reduction in the post-procedural increase in Cr [100].

Therefore, it remains inconclusive whether ACEI/ARBs increase or decrease the incidence of CIN and, currently, withholding RAAS blockers before CM administration is not recommended in guidelines [51,101]. Additional large-scale studies concerning type and dose of ACEI/ARB, ethnicity, and chronic/new users are needed to determine how to use ACEI/ARB in patients undergoing CM-using procedures.

In diabetic patients, metformin is widely prescribed as the first-line therapy. Metformin is mainly excreted by the kidneys and confers an increased risk of lactic acidosis when CIN occurs, although it does not increase the risk of CIN. However, as the reported incidence of metformin-associated lactic acidosis has been very low (<10 cases per 100,000 patient-years) [102], guidelines have become less strict. Based on recent U.S. Food and Drug Administration, ACR, and Radiological Society of the Netherlands guidelines, the CMSC recommends to stop taking metformin at the time of CM administration in (1) patients with eGFR < 30 mL/min/1.73 m^2^ receiving IV CM or IA CM with second pass renal exposure, (2) patients receiving IA CM with first pass renal exposure, and (3) patients with AKI. They also recommend to measure eGFR within 48 h and restart metformin if renal function has not changed significantly [51,103,104].

Minimizing the total volume of CM and using the least nephrotoxic CM should be applied in all cases. There have been efforts to reduce the contrast volume (iodine dose) as low as reasonably achievable during both CAG and CECT [50,105]. Clinicians should also consider the interval of CM administration when repeated procedures are needed because multiple doses of CM within a short period of time (24–72 h) increase the risk of CIN [10,106]. 

In addition to modifying patient-related risk factors and properly choosing the type and volume of CM, intravenous fluid hydration is the mainstay of CIN preventive strategies. Hydration is theoretically reasonable because it can correct or improve the patient’s volume status, dilute CM concentration, and increase kidney blood flow and tubular urine flow, which can subsequently reduce CM retention and toxic effects in the tubular lumen. Guidelines recommend intravascular volume expansion as CIN prophylaxis, but there is no consensus on the optimal hydration regime [8,51,92]. To achieve optimal hydration status without increasing the risk of pulmonary edema, two tailored hydration regimens have been proposed and widely investigated: (1) left ventricular end-diastolic pressure (LVEDP)-guided hydration and (2) urine flow rate (UFR)-guided hydration using the RenalGuard system.

The POSEIDON (Prevention of Contrast Renal Injury with Different Hydration Strategies) trial compared LVEDP-guided hydration with standard hydration in 396 patients undergoing cardiac catheterization [107]. All patients received a bolus infusion of normal saline (3 mL/kg) for 1 h prior to the procedure. During and for 4 h after the procedure, the LVEDP-guided group received normal saline at a rate of 1.5 to 5 mL/kg/h, depending on the LVEDP, and the control group received 1.5 mL/kg/h of normal saline. The total hydration volume was higher in the LVEDP-guided group (mean volume, 1727 mL vs. 812 mL, *p* < 0.001), and significantly fewer cases of CIN occurred in that group (6.7% vs. 16.3%, *p* = 0.005). Noticeably, the odds of CIN decreased by 9% for every additional 100 mL of normal saline administered (OR 0.91, 95% CI 0.89–0.94, *p* = 0.01). The reported rate of shortness of breath was 1.5% and similar in the two groups. The 6-month composite outcome that considered all-cause mortality, myocardial infarction, and renal replacement therapy (RRT) was better in the LVEDP-guided group than in the control group.

Increasing the UFR above 150 mL/h was reported to reduce CIN [108]. Theoretically, a high UFR will rapidly remove CM from the kidney, reducing its toxicity within the nephron. For UFR-guided hydration, a bolus of normal saline hydration plus IV furosemide (0.25 mg/kg) was initially administered to achieve UFR ≥300 mL/h, followed by urine output-matched hydration using the RenalGuard system to maintain that high UFR during and after contrast exposure [109,110]. With the RenalGuard system, no significant electrolyte imbalance or pulmonary edema was documented [111]. In a meta-analysis of six RCTs, the RenalGuard system was demonstrated to reduce CIN significantly in patients undergoing PCI [112].

Briguori et al. compared these two tailored hydration regimens in an RCT with 708 patients scheduled for coronary or peripheral angiography or angioplasty [113]. The total hydration volume was significantly higher in the UFR-guided group than in the LVEDP-guided group and UFR-guided hydration was superior to LVEDP-guided hydration in preventing CIN and 1-month major adverse events. In addition, acute pulmonary edema developed less often in the UFR-guided group than the LVEDP-guided group, although that difference was not significant. However, hypokalemia developed more often with UFR-guided hydration than LVEDP-guided hydration (6.2% vs. 2.3%, *p* = 0.013), and three (0.8%) patients experienced complications related to foley catheter insertion in the RenalGuard system.

Both preventive regimens mentioned above require relatively invasive procedures, and they were investigated mostly during coronary intervention or transcatheter aortic valve implantation. Recently, non-invasive methods to guide hydration have been reported.

The HYDRA study by Maioli et al. evaluated the effect of bioimpedance vector analysis (BIVA) to determine IV infusion volumes [114]. Three hundred and three patients with low body fluid levels as assessed by BIVA and scheduled for CAG were divided into two groups: the standard volume saline group (1 mL/kg/h for 12 h before and after the procedure) and the double volume saline group (2 mL/kg/h). As expected, significantly more patients in the double volume saline group achieved the optimal BIVA before the angiographic procedure (50.0% vs. 27.7%, *p* = 0.0001), and they showed a significantly lower incidence of CIN (11.5% vs. 22.3%, *p* = 0.015) than the standard volume saline group. In addition, the occurrence of CIN was lower (9.4%, 66 of 704) in patients with an optimal BIVA level on admission who were included in a registry group and received standard volume saline.

Yan et al. used inferior vena cava ultrasonography (IVCU) to guide hydration in chronic heart failure patients with New York Heart Association functional classification ≥2 and left ventricular ejection fraction <50% [115]. Two hundred and seven patients receiving CAG or PCI were divided into two groups: the control group (isotonic saline at a rate of 0.5 mL/kg/h for 6 h before and 12 h after the procedure) and the IVCU-guided hydration group (isotonic saline at a rate of 0.5, 1.0, or 1.5 mL/kg/h when their IVC diameter was >25, 20–25, or <20 mm, respectively, for the same time period). The hydration volume was significantly higher in the IVCU-guided group than the control group, and the incidence of CIN was significantly lower (12.5% vs. 29.1%, *p* = 0.004). Additionally, major adverse cardiovascular or cerebrovascular events during the 18-month follow-up occurred less often in patients who received IVCU-guided hydration.

These results indicate that a patient’s hydration status is a crucial factor associated with the development of CIN, and a sufficient volume expansion within a safe range under various guidance methods could be an important preventive strategy. As a non-invasive and cost-effective hydration method, oral hydration has been compared with IV hydration. Oral hydration can suppress the release of vasopressin and lead to rapid diuresis [116]. It has been shown to be non-inferior to IV hydration in preventing CIN [117,118]. A recently published NICIR study by Sebastia et al. was a phase III non-inferiority study comparing oral hydration with IV hydration in patients with CKD stage IIIb who underwent elective CECT [119]. The method of oral hydration was 500 mL of water 2 h before and 2000 mL in the following 24 h after CECT, which was the same method proposed by Kong et al. in patients undergoing CAG or PCI [120]. IV hydration used sodium bicarbonate (166 mmol/L) at 3 mL/kg/h starting 1 h before the procedure and 1 mL/kg/h during the hour after CECT. Oral hydration was shown to be non-inferior to IV hydration regarding the incidence of CIN, but baseline eGFR was significantly higher in the oral hydration group (39.0 vs. 36.0 mL/min/1.73 m^2^, *p* = 0.002) due to non-stratified randomization. 

Although hydration is the only evidence-based recommendation for the prevention of CIN, few studies have compared the incidence of CIN with and without hydration. The AMACING trial investigated the prophylactic value of hydration in 660 high-risk patients undergoing an elective procedure requiring CM administration [121]. The incidence of CIN was 2.6% (8 of 307) in the non-hydrated patients and 2.7% (8 of 296) in the hydrated patients, which was inconclusive evidence for the effectiveness of IV hydration. Although the results of the AMACING study led researchers to revisit previous studies assessing hydration versus no hydration, a meta-analysis by Jiang et al. of six RCTs with different hydration regimens reported that patients who received prophylactic hydration had a lower risk of CIN than those who did not [122]. In a subgroup analysis, they found that hydration offered no benefit to patients with a baseline eGFR of 30–60 mL/min/1.73 m^2^, which possibly reflected patients’ baseline hydration status. A more recent meta-analysis by Michel et al. analyzed 37 RCTs with 12,166 patients to assess IV volume expansion strategies [123]. Again, IV volume expansion was associated with a lower risk of CIN compared with no fluid administration or oral fluid intake. Furthermore, intensive IV volume expansion with an average absolute volume of 1.6 L over a 17 h peri-contrast exposure was associated with a reduced risk of CIN compared with standard volume expansion strategies. In the AMACING trial, a minimum volume of pre-warmed (37 °C) iopromide (300 mg iodine per mL) was used in all patients, with mean CM volumes of 92 and 89 mL in the hydration and no hydration groups, respectively. That might have contributed to the low incidence of CIN and explain the finding of no efficacy for hydration in high-risk patients. It also stresses the importance of minimizing contrast volume.

Cai et al. reviewed hydration strategies in 60 RCTs and performed a network meta-analysis to find an optimal strategy [124]. They reported that the RenalGuard system was best, followed by hemodynamic guidance monitoring for hydration. The latter reflected only three RCTs using central venous pressure, LVEDP, and bioimpedance.

With regard to the type of hydration, normal saline (0.9% sodium chloride) is recommended in the guidelines as the primary choice. Another type of fluid, apart from sodium bicarbonate, was investigated recently. Park et al. conducted a multicenter RCT to determine the efficacy of a balanced salt solution versus normal saline in high-risk patients undergoing scheduled CECT [125]. However, that study failed to meet its target enrollment and reported no significant differences between the two fluid groups containing a total of 493 patients. No optimal hydration strategy has been established as a preventive measure for CIN. Therefore, further subspecialized studies that consider both patient-related and procedure-related factors are needed to offer individually optimized volume expansion.

Due to concerns about CIN furthering renal damage, particularly in patients with advanced CKD (stage 4 or 5) who are not on maintenance dialysis, prophylactic hemodialysis or hemofiltration has been applied to remove CM. A meta-analysis by Cruz et al. in 2012 that included nine RCTs and two non-RCTs with 1010 patients (eight studies using hemodialysis and three using hemofiltration or hemodiafiltration) demonstrated no benefit of periprocedural RRT compared to standard medical therapy (RR 1.02, 95% CI 0.54–1.93), and hemodialysis appeared to actually increase the incidence of CIN (RR 1.61, 95% CI 1.13–2.28) [126]. With no favorable evidence of preventive RRT, current guidelines do not recommend using prophylactic hemodialysis or hemofiltration for the purpose of CIN prevention, regardless of renal function [12,51,92]. In addition, for patients on maintenance dialysis, neither extra hemodialysis nor the change in hemodialysis schedule in relation to CM administration is recommended, unless there is the risk of volume overload [51,92,93]. Patients on maintenance dialysis who have residual renal function (urine > 100 mL/day) should be treated as patients with advanced CKD who are not undergoing dialysis [8].

Nonetheless, studies on prophylactic hemofiltration against CIN have been conducted. Two studies by Marenzi et al. in 2003 and 2006 [127,128], included in the above described meta-analysis [126], showed that periprocedural hemofiltration decreased the incidence of CIN in CKD patients undergoing coronary interventions compared to saline hydration. Then, a study by Choi et al. in 2014 compared periprocedural versus simultaneous hemofiltration in CKD patients undergoing CAG and demonstrated better late-stage (days 5–30) renal outcome in the simultaneous hemofiltration group compared to the periprocedural hemofiltration group [129]. A pilot study was published in 2020 that investigated the protective effect of high flow-volume intermittent hemodiafiltration against CIN compared to saline hydration [130]. This novel technique with increased CM removal efficiency was applied just before and for 2.5 h after CM-using interventions (CAG, PCI, or percutaneous peripheral intervention) in patients with advanced CKD (stage 3b or 4) and reduced the incidence of CIN both at day 2–3 and 1 month compared to saline hydration. However, due to the invasiveness, bleeding risk, and costs, further studies are essential to provide sufficient evidence and to find a specific population who can benefit the most. At present, a careful risk–benefit assessment is needed in patients with advanced CKD who are not on maintenance dialysis. It is also important that vital diagnostic and interventional procedures requiring CM administration should not be withheld or postponed solely due to the risk of CIN in those patients.

As another preventive measure, scavenging of the ROS produced during CM administration was suggested, and IV sodium bicarbonate and oral N-acetylcysteine (NAC) have been widely tried for that purpose.

In the 2007 RENO study, 111 acute coronary syndrome patients undergoing emergency PCI were randomized [131]. The active prophylactic treatment group received 5 mL/kg/h of sodium bicarbonate solution plus 2400 mg of NAC in the same solution during 1 h preceding CM administration, and the fluid without NAC was continued at a rate of 1.5 mL/kg/h for 12 h after PCI. The control group received 1 mL/kg/h of isotonic saline for 12 h after PCI. Two 600 mg doses of oral NAC were administered the next day in both groups. Baseline characteristics, including kidney and heart function, were comparable in the two groups, but the occurrence of CIN was significantly lower in the active prophylactic treatment group than the control group (1.8% vs. 21.8%, *p* = 0.0009).

However, the PRESERVE trial showed different results. It was a large RCT using a 2-by-2 factorial design [132] and involved 5177 high-risk patients, scheduled for angiography, whose eGFR was 15–44.9 mL/min/1.73 m^2^ or 45–59.9 mL/min/1.73 m^2^ with diabetes. They received either IV 1.26% sodium bicarbonate or IV normal saline and either 5 days of 1200 mg NAC orally or an oral placebo. The trial demonstrated no benefit of IV sodium bicarbonate or oral NAC on the incidence of CIN and the 90-day composite outcome of death, need for dialysis, or persistent decline in kidney function.

The PRIMARY trial by Boccalandro et al. is a single center RCT of 382 CKD stage III–IV patients undergoing elective CAG to evaluate the 5-year outcomes of patients with CIN and to assess the long-term effects of hydration with sodium bicarbonate [20]. Patients who developed CIN had significantly higher 5-year mortality than those without CIN, but IV sodium bicarbonate showed no benefit over normal saline on the incidence of CIN, mortality, RRT, or major adverse kidney and cardiovascular events.

Another large RCT of 2308 patients that added 1200 mg of oral NAC to hydration (ACT trial) also showed that NAC offered no benefit in reducing the risk of CIN [133]. Therefore, no concrete evidence or consensus supports the routine use of either sodium bicarbonate or NAC. The conflicting result of the RENO study might be attributable to the dose and route of administration of the agents or factors related to the patients and procedures. In fact, the contrast volume used in the RENO study was much higher (mean volume of 290 and 279 mL in each group) than that in the PRESERVE trial (median volume of 85 mL in both groups) or PRIMARY trial (mean volume of 156 and 160 mL in each group).

Various pharmacologic strategies for preventing CIN have been evaluated, but the results have often conflicted with one another. Su et al. reviewed 150 RCTs that evaluated pharmaceutical agents in combination with hydration and classified the agents into 12 categories based on drug species or dose as follows: (1) natriuretic peptides: atrial natriuretic peptide, B-type natriuretic peptide, and carperitide; (2) vitamins and analogues: ascorbic acid, tocopherol, and α-lipoic acid; (3) high-dose statins: simvastatin (40–80 mg), rosuvastatin (20–40 mg), and atorvastatin (40–80 mg); (4) low-dose statins: simvastatin (10–20 mg), rosuvastatin (10 mg), and atorvastatin (10–20 mg); (5) prostaglandins: iloprost, alprostadil, misoprostol, and prostaglandin E1; (6) theophylline (aminophylline); (7) NAC; (8) fenoldopam; (9) sodium bicarbonate; (10) sodium bicarbonate plus NAC; (11) high-dose statins plus NAC; (12) hydration [134]. They assessed those 12 interventions using a Bayesian network meta-analysis and found that the use of high-dose statins plus NAC and high-dose statins on their own, both in combination with hydration, were the best and the second-best strategies for reducing CIN, respectively.

A meta-analysis by Ma et al. of 107 studies with 21,450 patients also demonstrated that the use of statins plus NAC plus saline hydration was the most effective strategy for preventing CIN in patients undergoing CAG [135].

Statins have pleiotropic effects, including causing improvements in vascular tone by increasing endothelial NO production and antiinflammatory and antioxidant effects that can contribute to renoprotection in CIN [136,137,138]. A meta-analysis by Zhou et al. of seven RCTs with 4256 patients demonstrated that short-term moderate or high-dose statin pretreatment reduced the occurrence of CIN [139]. Of note, the subgroup analysis in that study revealed that statin pretreatment exhibited a preventive effect in patients with both CKD and diabetes, but it did not reduce the risk of CIN in non-diabetic patients with CKD. Both atorvastatin and rosuvastatin showed protective effects against CIN in patients with CKD, but one study using a high dose of simvastatin showed no preventive effect on CIN [140].

However, most of the patients included in the studies using statins received CAG or cardiac catheterization, and, in those patients, statins might reduce the incidence of CIN via their beneficial effects on underlying vascular disease, including coronary artery disease. For statins to be generally recommended as a preventive measure for CIN, further studies of patients with different underlying diseases and procedures are needed.

In summary, both the ESUR and KDIGO guidelines currently recommend IV volume expansion with either saline or sodium bicarbonate solutions in patients at risk of CIN, although no benefit of IV sodium bicarbonate has been demonstrated over normal saline, and IV saline hydration is preferred. Neither guideline recommends oral hydration as the sole preventive method [51,92]. In addition, they make no recommendations for pharmacological prophylaxis because the preventive effect of pharmaceutical agents has not been consistently and fully validated, although the KDIGO guideline does suggest using oral NAC with IV hydration in patients at risk of CIN with a very low grade of evidence (2D) [92].

## 6. Conclusions

Patients at risk of CIN should be carefully identified before procedures requiring CM. Patients exposed to multiple risk factors for CIN including accompanied comorbidities such as impaired renal function, heart failure, and diabetes, the amount of contrast volume, and hypotension are at increased risk. Some risk scoring systems such as Mehran’s score are available for patients undergoing CAG or PCI. Modifiable risk factors, both patient- and procedure-related, should be corrected. It is advisable to withhold nonessential nephrotoxic medications in at-risk patients. Guidelines recommend using IOCM or LOCM with efforts to lower the CM volume. However, minimizing contrast volume should not decrease diagnostic accuracy, even in patients with advanced CKD. Currently, hydration is the only evidence-based method for CIN prevention and normal saline is preferred. As renal impairment is the most important risk factor of CIN, prophylaxis with saline hydration is guided based on a patient’s eGFR. Patients with a stable baseline eGFR ≥ 45 mL/min/1.73 m^2^ are generally considered to be safe from CIN. Although the threshold of eGFR for prophylactic hydration is different according to guidelines and the ideal hydration regime is uncertain, the basic principle is to provide sufficient volume expansion without increasing the risk of pulmonary edema. The typical hydration volume is 1–3 mL/kg/h or a fixed volume (e.g., 500 mL normal saline) and the duration is 1–4 h before and 3–12 h after CM administration. The hydration regimen should be individualized, particularly in patients at risk of volume overload, and an assessment of patient’s volume status before CM administration is needed. Statins with or without NAC, combined with hydration, may mitigate the risk of CIN in patients undergoing coronary intervention, but further studies are required to recommend it. In the future, novel pharmaceutical agents targeting the pathogenic signaling pathways of CIN should be developed and validated in large-scale clinical trials to reverse the course of CIN, which is the ultimate goal for CIN management. Moreover, the accumulation of data from studies that take into account individual characteristics and risk factors will be able to provide detailed preventive strategies against CIN. 

## Figures and Tables

**Figure 1 diagnostics-12-00180-f001:**
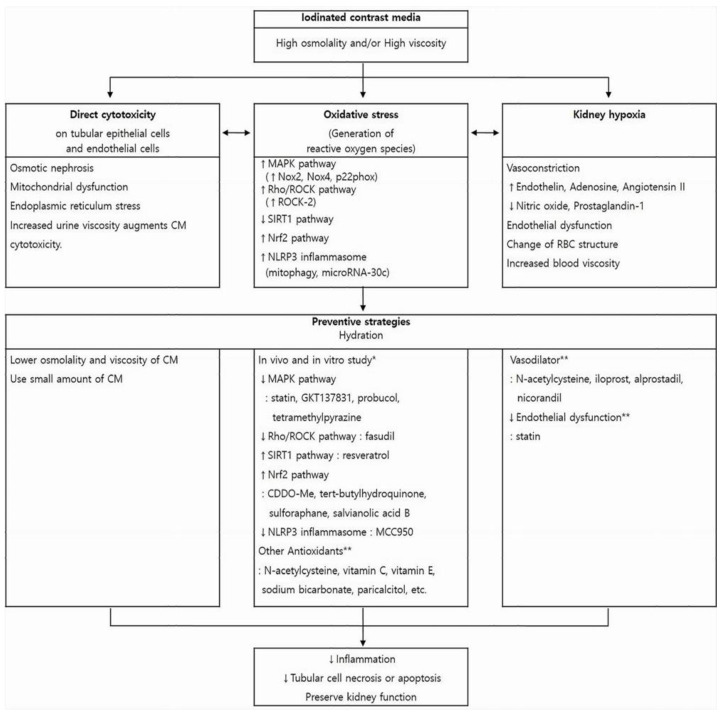
Pathophysiology of contrast-induced nephropathy (CIN) and promising strategies to preserve kidney function. Iodinated CM has direct cytotoxic effect on endothelial cells and renal tubular epithelial cells, induces vasoconstriction causing hypoxia in the outer medulla, and enhances the generation of reactive oxygen species. These changes influence one another and ultimately lead to kidney injury. Each box contains underlying mechanisms relevant to those three pathways. Hydration is the mainstay of CIN preventive strategies and can reduce harmful effect of CM in all three aspects. Other previously reported preventive measures and pharmaceutical agents are presented with regard to each pathogenic process. * These pharmaceutical agents have been studied in in vitro and in vivo experiments to reduce oxidative stress, that is, to reverse each pathogenic pathway. However, because Nrf2 expression increases during CM-induced oxidative stress as a cytoprotective response, Nrf2 activation is preventive against CIN. ** The preventive role of these agents on CIN is controversial. CM, contrast media; MAPK, mitogen-activated protein kinase; Nox, nicotinamide adenine dinucleotide phosphate oxidase; ROCK, rho-kinase; SIRT1, silent information regulator 1; Nrf2, nuclear factor erythroid 2-related factor 2; RBC, red blood cell.

**Table 1 diagnostics-12-00180-t001:** Risk factors predisposing the development of contrast-induced nephropathy. Risk factors for contrast-induced nephropathy (CIN) can be divided into patient-related and procedure-related risk factors. Some patient-related risk factors such as volume depletion and using nephrotoxic medications are modifiable. With regard to procedure-related risk factors, the risk of CIN varies according to type, volume, and route of CM administration. Atheroembolism related to catheter manipulation and repeated CM administration also poses an increased risk of CIN. CM, contrast media.

Patient-Related	Impaired renal function Diabetes mellitus Effective intravascular volume depletion: dehydration, blood loss, congestive heart failure, liver cirrhosis, nephrosis Advanced age Female gender Cardiovascular disease including hypertension Malignancy Inflammation Anemia Hyperuricemia Nephrotoxic medications: diuretics, nonsteroidal antiinflammatory drugs, aminoglycosides, amphotericin B, antiviral drugs such as acyclovir, cyclosporine A, cisplatin
Procedure-Related	Route of CM administration: intra-arterial vs. intravenous administration Type of procedure: catheter-based procedure Type of CM Volume of CM Repeated CM administration within 24–72 h

**Table 2 diagnostics-12-00180-t002:** Strategies to reduce the development of contrast-induced nephropathy. Preventive strategies against contrast-induced nephropathy (CIN) are presented, taking into account patient- and procedure-related risk factors and CIN pathophysiology. * Hydration is a patient-, procedure-, and pathophysiology-related preventive strategy against CIN.

Patient-related	Risk stratification of individual patients Evaluate and correct patient’s volume status Correct modifiable factors including cessation of nephrotoxic drugs
Procedure-related	Use low-osmolar or iso-osmolar contrast media Minimize the volume of contrast media - limit maximum contrast volume - consider the interval of contrast administration
Pathophysiology-related	Hydration * Pharmaceutical agents targeting pathogenic process including oxidative stress

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
