# Peer review of "The Pathophysiology and the Management of Radiocontrast-Induced Nephropathy"

_diagnostics, 2022, doi:10.3390/diagnostics12010180_

Round 1

Reviewer 1 Report

At the beginning of the article, It is needed to describe a methodology of the review- in which data base the authors were looking for the information, what kind of information did they look for, which were the key words, which are the inclusion and and the exclusion criteria for the studies that they found, the number of the studies that the found etc .

Author Response

  • We appreciate your comments. As you recommended, we added the method how we looked for articles at the end of the introduction section.

Line 36-44

For this review, literature search was performed using electronic databases such as PubMed and Embase. The search strategy included any original article or review about contrast induced nephropathy. We used combinations of the following search terms: contrast, iodinated contrast media, nephropathy, risk, score, incidence, guideline, definition, intravenous, intra-arterial, biomarker, chronic kidney disease, diabetes, metformin, angiotensin, pathophysiology, oxidative stress, Rho, ROCK, sirtuin, SIRT1, Nrf2, NLRP3 inflammasome, prevention, hydration, RenalGuard system, sodium bicarbonate, N-acetylcysteine, statin. Both in vivo or vitro experimental studies and clinical studies were reviewed.

Reviewer 2 Report

Major comments:

This manuscript is a review on the pathophysiology and management of contrast-induced nephropathy. The latest findings from the field are given, some parts of the manuscript are well systematized - part on pathophysiology, on risk factors (schematically well presented without patient risk stratification), preventive strategies, but there is no concise conclusion with a recommendation for everyday clinical practice.

In the conclusion, the authors mentioned high-risk patients, and in the rest of the manuscript patients at increased risk. Nowhere in the manuscript is it stated which patients are at high risk and on the basis of which and how many parameters / characteristics of patients they are so stratified. Patients at increased risk may be high or moderate / low risk, and thus their prevention of CIN varies in certain segments. This segment of the manuscript needs to be refined.

Taking into account all the ambiguities, controversies and current evidence (described and undescribed in this manuscript), it remains unclear in the manuscript, especially in the conclusion, how to approach a high-risk patient, what risk stratification and what preventive measures to use in everyday clinical practice. On the one hand, an individualized approach is stated, and on the other hand, a very general answer is offered as an approach in the conclusion. To overcome these two extremes, it is necessary to summarize whether there are differences in risk management approach and to propose and describe your algorithm of approach to the patient with regard to patient risk (low / moderate / high risk) and eGFR before applying contrast, with additional remarks and indication of possible differences in prevention depending on the route of CM administration (IV vs AI), its dose and volume, and the type of imaging method / intervention used (CECT vs CAG vs DSA, PCI vs non coronary PTA…) with reference to the current recommendations and recent findings.

In addition, does the new insights you cite in the manuscript change the current approach to CIN prevention?

It should also be noted that the management of CIN in this manuscript refers primarily to the adult population. Indicate this in the manuscript. Otherwise, a section on CIN prevention in children should be added.

In addition, there is no review of 1) the use and discontinuation of metformin nor ACEi and sartan in the prevention of CIN. This should definitely be commented on because these are commonly prescribed medications (if indicated, when and for how long; if not, why not); 2) the role of dialysis in the prevention of CIN in patients with CKD G4 and G5 who are not yet on dialysis as well as those on dialysis with and without residual urine (Is dialysis indicated at all? What are the recommendations and level of evidence for the role of dialysis in the prevention of CIN in these groups?)

Correction of figures and tables is required. The authors in Figure 1 list the use of herbs as part of CIN prevention strategies. Which herbs does it apply to? This cannot remain so without further clarification below the Figure. Namely, one should be very careful due to herbal nephropathy (e.g. CHN i.e. AAN etc.)

Minor comments:

Correction of figures and tables is required.

Each table and figure should be clear and understandable to the reader regardless of the textual part of the article. All abbreviations given in the tables and / or figures should be listed below them, stating the meaning of the abbreviations used.

In Figure 1, in the iodinated contrast media box, instead of high osmolality, high viscosity, write high osmolality AND / OR high viscosity, which would be more in line with what you state in detail in the manuscript.

Table 1. - ex. - example? …

Author Response

Major comments:

This manuscript is a review on the pathophysiology and management of contrast-induced nephropathy. The latest findings from the field are given, some parts of the manuscript are well systematized - part on pathophysiology, on risk factors (schematically well presentedwithout patient risk stratification), preventive strategies, but there is no concise conclusion with a recommendation for everyday clinical practice.In the conclusion, the authors mentioned high-risk patients, and in the rest of the manuscript patients at increased risk. Nowhere in the manuscript is it stated which patients are at high risk and on the basis of which and how many parameters / characteristics of patients they are so stratified. Patients at increased risk may be high or moderate / low risk, and thus their prevention of CIN varies in certain segments. This segment of the manuscript needs to be refined.

  • We appreciate reviewer’s precise comments. As reviewer mentioned, recommendations for clinical practice and details of risk stratification previously produced especially for coronary intervention were reinforced. And the words ‘high risk patients’ were also switched to ‘patients at risk of CIN’ or ‘at-risk patients’.

Line 650-676

Patients at risk of CIN should be carefully identified before procedures requiring CM. Patients exposed to multiple risk factors for CIN including accompained comorbidities such as impaired renal function, heart failure and diabetes, the amount of contrast volume and hypotension are at increased risk. Some risk scoring systems such as Mehran’s score are available for patients undergoing CAG or PCI. Modifiable risk factors, both patient- and procedure-related, should be corrected. It is advisable to withhold nonessential nephrotoxic medications in at-risk patients. Guidelines recommend using IOCM or LOCM with efforts to lower the CM volume. However, minimizing contrast volume should not decline diagnostic accuracy, even in patients with advanced CKD. Currently, hydration is the only evidence-based method for CIN prevention and normal saline is preferred. As renal impairment is the most important risk factor of CIN, prophylaxis with saline hydration is guided based on a patient’s eGFR. Patients with a stable baseline eGFR ≥45 ml/min/1.73 m2 are generally considered to be safe from CIN. Although the threshold of eGFR for prophylactic hydration is different according to guidelines and the ideal hydration regime is uncertain, the basic principle is to provide sufficient volume expansion without increasing the risk of pulmonary edema. Typical hydration volume is 1-3 ml/kg/h or fixed volume (e.g., 500mL normal saline) and duration is 1-4h before and 3-12h after CM administration. Hydration regimen should be individualized, particularly in patients at risk of volume overload, and assessment of patient’s volume status before CM administration is needed. Statins with or without NAC, combined with hydration, may mitigate the risk of CIN in patients undergoing coronary intervention, but further studies are required to recommend it. In the future, novel pharmaceutical agents targeting the pathogenic signaling pathways of CIN should be developed and validated in large-scale clinical trials to reverse the course of CIN, which is the ultimate goal for CIN management. Also, accumulation of data from studies that take into account individual characteristics and risk factors will be able to provide detailed preventive strategies against CIN.

Line 195-200

Mehran first developed a risk scoring system, which involves 8 clinical and procedural variables such as age over 75 years, existence of decrease of renal function, hypotension, congestive heart failure, diabetes, and anemia, use of intra-aortic balloon pump (IABP) and large contrast volume, to predict CIN after PCI. More than half of patients having higher score more than 16 were reported to experience CIN. Exposure to more risk factors is valuable to define patients at-risk for CIN.

Taking into account all the ambiguities, controversies and current evidence (described and undescribed in this manuscript), it remains unclear in the manuscript, especially in the conclusion, how to approach a high-risk patient, what risk stratification and what preventive measures to use in everyday clinical practice. On the one hand, an individualized approach is stated, and on the other hand, a very general answer is offered as an approach in the conclusion. To overcome these two extremes, it is necessary to summarize whether there are differences in risk management approach and to propose and describe your algorithm of approach to the patient with regard to patient risk (low / moderate / high risk) and eGFR before applying contrast, with additional remarks and indication of possible differences in prevention depending on the route of CM administration (IV vs AI), its dose and volume, and the type of imaging method / intervention used (CECT vs CAG vs DSA, PCI vs non coronary PTA…) with reference to the current recommendations and recent findings.

  • We appreciate reviewer’s precise comments. As we mentioned above, we revised the conclusion more comprehensively, comprising the findings in the manuscript and based on guidelines.
  • Line 650-676
  • Patients at risk of CIN should be carefully identified before procedures requiring CM. Patients exposed to multiple risk factors for CIN including accompained comorbidities such as impaired renal function, heart failure and diabetes, the amount of contrast volume and hypotension are at increased risk. Some risk scoring systems such as Mehran’s score are available for patients undergoing CAG or PCI. Modifiable risk factors, both patient- and procedure-related, should be corrected. It is advisable to withhold nonessential nephrotoxic medications in at-risk patients. Guidelines recommend using IOCM or LOCM with efforts to lower the CM volume. However, minimizing contrast volume should not decline diagnostic accuracy, even in patients with advanced CKD. Currently, hydration is the only evidence-based method for CIN prevention and normal saline is preferred. As renal impairment is the most important risk factor of CIN, prophylaxis with saline hydration is guided based on a patient’s eGFR. Patients with a stable baseline eGFR ≥45 ml/min/1.73 m2 are generally considered to be safe from CIN. Although the threshold of eGFR for prophylactic hydration is different according to guidelines and the ideal hydration regime is uncertain, the basic principle is to provide sufficient volume expansion without increasing the risk of pulmonary edema. Typical hydration volume is 1-3 ml/kg/h or fixed volume (e.g., 500mL normal saline) and duration is 1-4h before and 3-12h after CM administration. Hydration regimen should be individualized, particularly in patients at risk of volume overload, and assessment of patient’s volume status before CM administration is needed. Statins with or without NAC, combined with hydration, may mitigate the risk of CIN in patients undergoing coronary intervention, but further studies are required to recommend it. In the future, novel pharmaceutical agents targeting the pathogenic signaling pathways of CIN should be developed and validated in large-scale clinical trials to reverse the course of CIN, which is the ultimate goal for CIN management. Also, accumulation of data from studies that take into account individual characteristics and risk factors will be able to provide detailed preventive strategies against CIN.

In addition, does the new insights you cite in the manuscript change the current approach to CIN prevention?

  • Although it is not that new that hydration is the only proven preventive strategy against CIN at present, we tried to emphasize the importance of assessing patient’s volume status and providing sufficient fluid volume (to make optimal volume status unless it causes pulmonary edema) with possible methods such as Renalguard system or IVCU. And we also tried to summarize potential medications or methods that may be implemented in the future.

It should also be noted that the management of CIN in this manuscript refers primarily to the adult population. Indicate this in the manuscript. Otherwise, a section on CIN prevention in children should be added.

  • According to reviewer’s comments, we added following sentences as you recommended.

Line 335-337

Herein, we review preventive strategies studied in adult patients undergoing procedures using CM. Therefore, the management of CIN in this article refers primarily to the adult population.

In addition, there is no review of 1) the use and discontinuation of metformin nor ACEi and sartan in the prevention of CIN. This should definitely be commented on because these are commonly prescribed medications (if indicated, when and for how long; if not, why not); 2) the role of dialysis in the prevention of CIN in patients with CKD G4 and G5 who are not yet on dialysis as well as those on dialysis with and without residual urine (Is dialysis indicated at all? What are the recommendations and level of evidence for the role of dialysis in the prevention of CIN in these groups?)

  • We appreciate reviewer’s precise comments. As reviewer mentioned, the review of guidelines and studies about 1) the use of metformin and ACEi/ARB and 2) the role of dialysis in CKD patients were added.

Line 357-407

When clinically feasible, it is recommended to withhold nonessential nephrotoxic medications before CM administration, those are listed in Table 1 [51,92,93]. Renin-angiotensin-aldosterone system (RAAS) blockers [angiotensin-converting enzyme inhibitors (ACEI) and angiotensin receptor blockers (ARB)] are generally used in patients with cardiovascular disease, CKD, and diabetes. Because RAAS blockade can change renal hemodynamics and induce AKI, the effect of ACEI/ARBs on the incidence of CIN is of great concern [94]. Wu et al. performed a meta-analysis with 14 studies composed of 15,447 patients (7,288 treated with ACEIs or ARBs and 8,159 in the control group) undergoing CAG [95]. The overall estimate demonstrated significantly increased risk of CIN in the ACEI/ARBs group compared to the control group (OR 1.50, 95%CI 1.03-2.18, P = 0.03), but the association was not observed in the 7 RCTs (OR 0.88, 95%CI 0.41-1.90, P=0.74). In a recent meta-analysis by Want et al. that included 12 studies with 14 trials, containing 4,864 patients (2,484 treated with RAAS blockers and 2,380 in the control group), the pooled relative risk of CIN incidence in the RAAS blocker group was 1.22 (95% CI 0.81-1.84) [96]. However, an increased risk of CIN in the RAAS blocker group was observed among older people (RR 2.02, 95% CI 1.21-3.36), non-Asians (RR 2.30, 95% CI 1.41-3.76), chronic users (RR 1.69, 95% CI 1.10-2.59), and studies with larger sample size (population ≥200, RR 1.83, 95% CI 1.28-2.63).

Among studies included in above described meta-analyses, only a few RCTs directly investigated the effects of withholding ACEI/ARBs on the incidence of CIN. Discontinuing captopril 36h before PCI did not change the incidence of CIN in patients with sCr ≤1.5 mg/dL or GFR ≥60 ml/min [97]. Withholding ACEI/ARBs 24h before CAG did not appear to influence the incidence of CIN in patients with CKD stages 3–4 [98]. Recently, Motes et al. performed a retrospective study and analyzed changes of renal function during one-month post CAG in CKD stages 2-5 patients who take ACEI/ARB and are not on dialysis [94]. This study revealed that the continuation of ACEI/ARB was not associated with significant renal injury after CAG. However, post-hoc analysis of an RCT by Wolak et al. showed that the continuation of ACEI/ARB was associated with a significant decrease in eGFR 48h post CAG in patients with baseline eGFR <60 ml/min compared to the discontinuation group, while there was no significant difference in changes of renal function between the two groups in patients with eGFR ≥60 ml/min [99]. Likewise, in patients with moderate renal insufficiency (Cr ≥1.7 mg/dL within 3 months and/or Cr ≥1.5 mg/dL within 1 week before cardiac catheterization), withholding ACEI/ARB resulted in a non-significant reduction in CIN and a significant reduction in the post-procedural increase of Cr [100].

Therefore, it remains inclusive whether ACEI/ARBs increase or decrease the incidence of CIN and currently, withholding RAAS blockers before CM administration is not recommended in guidelines [51,101]. Additional large scale studies concerning type and dose of ACEI/ARB, ethnicity, and chronic/new users are needed to determine how to use ACEI/ARB in patients undergoing CM-using procedures.

In diabetic patients, metformin is widely prescribed as the first-line therapy. Metformin is mainly excreted by the kidneys and confers an increased risk of lactic acidosis when CIN occurs, although it does not increase risk of CIN. However, as the reported incidence of metformin-associated lactic acidosis has been very low (<10 cases per 100,000 patient-years) [102], guidelines have become less strict. Based on recent U.S. Food and Drug Administration, ACR and Radiological Society of the Netherlands guidelines, CMSC recommends to stop taking metformin at the time of CM administration in 1) patients with eGFR<30 ml/min/1.73 m2 receiving IV CM or IA CM with second pass renal exposure, 2) patients receiving IA CM with first pass renal exposure, and 3) patients with AKI. They also recommend to measure eGFR within 48h and restart metformin if renal function has not changed significantly [51,103,104]

Line 537-571

Due to concerns about CIN furthering renal damage particularly in patients with advanced CKD (stage 4 or 5) who are not on maintenance dialysis, prophylactic hemodialysis or hemofiltration has been applied to remove CM. A meta-analysis by Cruz et al. in 2012 that included 9 RCTs and 2 non-RCTs with 1,010 patients (8 studies using hemodialysis and 3 using hemofiltration or hemodiafiltration) demonstrated no benefit of periprocedural RRT compared to standard medical therapy (RR 1.02, 95% CI, 0.54-1.93) and hemodialysis appeared to actually increase the incidence of CIN (RR 1.61, 95% CI, 1.13-2.28)[127]. With no favorable evidence of preventive RRT, current guidelines do not recommend using prophylactic hemodialysis or hemofiltration for the purpose of CIN prevention, regardless of renal function [12,51,92]. In addition, for patients on maintenance dialysis, neither extra hemodialysis nor the change of hemodialysis schedule in relation to CM administration is recommended, unless there is risk of volume overload [51,92,93]. Patients on maintenance dialysis who have residual renal function (urine >100 ml/day) should be treated as patients with advanced CKD who are not undergoing dialysis[107].

Nonetheless, studies on prophylactic hemofiltration against CIN have been conducted. Two studies by Marenzi et al. in 2003 and 2006 [128,129], included in the above described meta-analysis[127], showed that periprocedural hemofiltration decreased the incidence of CIN in CKD patients undergoing coronary interventions compared to saline hydration. Then, a study by Choi et al. in 2014 compared periprocedural versus simultaneous hemofiltration in CKD patients undergoing CAG and demonstrated better late-stage (days 5–30) renal outcome in the simultaneous hemofiltration group compared to the periprocedural hemofiltration group [130]. A pilot study was published in 2020 that investigated the protective effect of high flow-volume intermittent hemodiafiltration against CIN compared to saline hydration [131]. This novel technique with increased CM removal efficiency was applied just before and until 2.5h after CM-using interventions (CAG, PCI, or percutaneous peripheral intervention) in patients with advanced CKD (stage 3b or 4) and reduced the incidence of CIN both at day 2–3 and 1 month compared to saline hydration. However, due to the invasiveness, bleeding risk, and costs, further studies are essential to provide sufficient evidence and to find specific population who can benefit the most. At present, a careful risk-benefit assessment is needed in patients with advanced CKD who are not on maintenance dialysis. It is also important that vital diagnostic and interventional procedures requiring CM administration should not be withheld or postponed solely due to the risk of CIN in those patients.

Correction of figures and tables is required. The authors in Figure 1 list the use of herbs as part of CIN prevention strategies. Which herbs does it apply to? This cannot remain so without further clarification below the Figure. Namely, one should be very careful due to herbal nephropathy (e.g. CHN i.e. AAN etc.)

  • We appreciate reviewer’s precise comments. As reviewer mentioned, the word, ‘herbs’ was removed (in other antioxidants part) due to the concern of misunderstanding and herbal nephropathy.

There were a few in vivo and in vitro experiments using herbal extracts (tanshinone IIA, 3-epi-iso-seco-tanapartholide isolated from Artemisia argyi, and xuezhikang) to find anti-oxidant effects against CIN, but, unlike other antioxidants presented in this part, they were not studied in humans.

Minor comments:

Correction of figures and tables is required.

Each table and figure should be clear and understandable to the reader regardless of the textual part of the article. All abbreviations given in the tables and / or figures should be listed below them, stating the meaning of the abbreviations used.

  • We appreciate reviewer’s precise comments. We added supplement explanations including the meaning of the abbreviations below every table and figure.

In Figure 1, in the iodinated contrast media box, instead of high osmolality, high viscosity, write high osmolality AND / OR high viscosity, which would be more in line with what you state in detail in the manuscript.

Table 1. - ex. - example? …

  • We corrected the figure 1 as reviewer mentioned. And ex. (for example) in table 1 was switched to ‘such as’.

Reviewer 3 Report

The authors present a very valuable work. Acute kidney injury related to examinations or interventions using contrast medium has a high clinical importance. The sections of the manuscript clearly discuss the problem, we obtain an insight into terminology of contrast medium-related renal pathologies. Furthermore, we can read a good, well-written description of risk factors, pathophysiology of development of renal impairment. In addition, authors discuss the management/prevention strategy. In summary, this work is an excellent, valuable summary, which can be accepted in tis present form.

Author Response

We deeply appreciate the encouraging comments and compliments.

Reviewer 4 Report

This is a comprehensive, well-written and structured narrative review  which focuses on pathophysiology and management of radiocontrast induced nephropathy. The authors performed an extensive literature search and the description of the pathophysiologic mechanisms is very detailed and updated.

Prevention and management are also analyzed extensively.

Most recent studies have been incorporated as well.

The paper fits well to the topic of the special issue and can be published in its current form.

Author Response

(The authors gave the same response as above.)

Round 2

Reviewer 1 Report

No comments.

Author Response

We appreciate your comments. 

Reviewer 2 Report

The manuscript has been significantly improved according to the suggestions and comments of the reviewer.
Minor correction is needed in Table 1. The title is misspelled - contrast induced without spaces. In addition, the term Table 1 should not be repeated below the table.

Author Response

We appreciate your comments. We corrected the misspelling and removed the repeated term 'Table 1' and 'Table 2' below each table.